# CEUS Bosniak Classification—Time for Differentiation and Change in Renal Cyst Surveillance

**DOI:** 10.3390/cancers15194709

**Published:** 2023-09-25

**Authors:** Kathleen Möller, Christian Jenssen, Jean Michel Correas, Ehsan Safai Zadeh, Michele Bertolotto, André Ignee, Yi Dong, Vito Cantisani, Christoph F. Dietrich

**Affiliations:** 1Medical Department I/Gastroenterology, Sana Hospital Lichtenberg, 10365 Berlin, Germany; 2Department of Internal Medicine, Krankenhaus Märkisch-Oderland, 15344 Strausberg, Germany; 3Brandenburg Institute of Clinical Medicine, Medical University Brandenburg, 16816 Neuruppin, Germany; 4Biomedical Imaging Laboratory, UMR 7371-U114, University of Paris, 75006 Paris, France; 5Department of Biomedical Imaging and Image-Guided Therapy, Medical University of Vienna, 1090 Vienna, Austria; 6Department of Radiology, Ospedale di Cattinara, University of Trieste, 34149 Trieste, Italy; 7Department of Medical Gastroenterology, Julius-Spital, 97070 Würzburg, Germany; 8Department of Ultrasound, Xinhua Hospital Affiliated to Shanghai Jiaotong University School of Medicine, Shanghai 200092, China; 9Department of Radiology, Oncology, and Anatomy Pathology, “Sapienza” University of Rome, 00185 Rome, Italy; 10Department Allgemeine Innere Medizin, Hirslanden Klinik Beau-Site, 3013 Bern, Switzerland

**Keywords:** renal cysts, Bosniak classification, CEUS, observation strategies

## Abstract

**Simple Summary:**

The present work deals with the different imaging procedures in the surveillance of renal cysts. The accuracy, interobserver variability, strengths and weaknesses of CT, MRI and CEUS in the classification of renal cysts are compared. The analysis refers to the original classification by MA Bosniak 1986, the update by GM Israel and MA Bosniak 2005, the version by SG Silverman et al. 2019, and the EFSUMB CEUS proposal by V Cantisani et al. 2021. The authors come to the conclusion that benign renal cysts, which are subject to surveillance, can and should be examined with contrast-enhanced ultrasound. CEUS is equivalent in the evaluation of renal cysts requiring surveillance compared with CE-CT and MRI. Appropriate qualification of the investigators is an unconditional requirement. MRI and CT can be used to investigate specific questions or for staging in case of surgery.

**Abstract:**

It is time for a change. CEUS is an established method that should be much more actively included in renal cyst monitoring strategies. This review compares the accuracies, strengths, and weaknesses of CEUS, CECT, and MRI in the classification of renal cysts. In order to avoid overstaging by CEUS, a further differentiation of classes IIF, III, and IV is required. A further development in the refinement of the CEUS-Bosniak classification aims to integrate CEUS more closely into the monitoring of renal cysts and to develop new and complex monitoring algorithms.

## 1. Introduction (Incidence, Imaging Modalities, Clinical Problems, Aim)

### 1.1. What Is the Prevalence of Renal Cystic Lesions? 

In ultrasound (US), computed tomography (CT), and magnetic resonance imaging (MRI) studies, the literature reports a wide range of prevalence of renal cysts (4–41%), depending on the patient spectrum and the sensitivity of the imaging modality [1]. The prevalence of renal cysts is highly dependent on age: for example, in a retrospective US study of 8551 patients, the prevalence was reported to be 0.6% in patients in the third decade of life and >30% in those over 80 years of age [2]. Among 10,261 people aged 24–63 years in a Korean health program, 5.4% had renal cysts on sonographic examination [3]. This represents a prevalence of 1.85% for those aged ≤40 years to 12.41% for of those aged >60 years [3]. In younger patients (<50 years old), simple renal cysts tend to increase in size more rapidly, but still do not result in aggressive disease [3]. In the MRI Study of Health in Pomerania [1], the prevalence, number, and mean size of renal cysts were higher in men (34%) compared to women (21%). The prevalence increased with age. Thus, in 15% of 20–29-year-old individuals and 55% of those over 70, renal cysts have been observed. 

### 1.2. Are There Specific Risk Factors for Development of Renal Cystic Lesions?

In addition to the genetic diseases associated with cystic kidney disease, male sex, older age, hypertension, proteinuria, renal stones, renal dysfunction, and smoking are significantly associated with increased renal cyst prevalence [1,3,4,5].

### 1.3. What Is the Risk of Renal Cystic Lesions Being Malignant Compared with Being Benign?

Depending on the histomorphological characteristics and pathogenetic factors, different levels of risk of malignant transformation have been described for renal cysts [6]. 

The risk of malignancy of simple cysts is reported to be <1% [7,8]. Case reports of simple cysts that have developed malignancy have been published [9,10]. In addition, cysts that develop in the setting of chronic renal disease have a higher risk of malignant degeneration [8].

### 1.4. Non-Neoplastic Cystic Lesions

Non-neoplastic renal cystic lesions include simple cysts, traumatic cysts, infectious cysts, and aneurysms [11,12,13,14]. The characteristics of these lesions have been explained and summarized in several studies [11,12,13,14].

### 1.5. Benign Neoplastic Cystic Lesions

The benign renal neoplastic cystic lesions include cystic nephroma, mixed epithelial and stromal tumor, cystic renal oncocytoma, angiomyolipoma with epithelial cysts, lymphangioma, angiomyolipoma with epithelial cysts, and cystic partially differentiated nephroblastoma [6,11,15,16]. In cystic nephroma, mixed epithelial and stromal tumor, and cystic renal oncocytoma, malignant transformation has been reported [16,17], whereas to date, no malignant transformation has been described in the other entities. However, due to the rarity of these types of cysts, the data are limited [16,18,19]. The characteristics of these lesions have been explained and summarized in several studies [6,11,15,16,17,18,19].

In a pathology study, 33% of renal cystic masses were benign. Of these, 91% were simple cysts and 9% were mixed epithelial and stromal tumors [20].

### 1.6. Malignant Neoplastic Cystic Lesions

Cystic variants of renal cell carcinomas (RCCs) and cystic variants of sarcomas are described as malignant renal cystic lesions [16,20,21,22,23]. The characteristics of these lesions have been explained and summarized in several studies [16,20,22,23].

In a pathology study, 67% of cystic masses were malignant. Of these, 63% were clear cell renal cell carcinomas and 25% were papillary renal cell carcinomas. Overall, 79% of the tumors were low-stage lesions (pT1) [20]. There was a trend toward malignant cysts in older patients [20]. Among 66% of malignant Bosniak III renal cysts, clear cell RCC accounted for 41%, papillary RCC for 10%, mixed RCC for 7%, and chromophobe RCC and multilocular RCC for 3% each [24]. 

### 1.7. What Are the Possible Symptoms and Complications of Renal Cystic Lesions?

Simple renal cysts do not usually cause any symptoms [25]. In the event of obstructive or displacing behavior of cysts, pain; urinary retention; and, in the case of infection of the cyst contents, fever and increased inflammatory parameters may be present [25,26,27].

### 1.8. Which Imaging Techniques Are Used for Examination of Cystic Renal Lesions?

In addition to CT as the primary method of choice, other imaging modalities, such as MRI, B-mode US, and contrast-enhanced US, are commonly used to evaluate renal cysts [7,28].

### 1.9. Summary of Existing Guidelines on Management of Cystic Renal Lesions

There are already several guidelines for the diagnosis and management of renal cystic lesions. We refer to the published literature [7,29,30,31,32]. Renal cysts are common, especially in the elderly. Most of them are benign. It is not understandable why predominantly potentially benign lesions should be monitored by CT or MRI. In addition, CT is associated with radiation exposure. These methods should be reserved for truly complex findings. The focus of the present work was to review, based on objective criteria, when it should be indicated to perform follow-up controls by CEUS. 

### 1.10. Aim

The aim of this analysis was to review and compare the accuracy, interobserver variability, strengths, and weaknesses of CT, MRI, and CEUS in the classification of renal cysts. The analysis focuses on the original classifications by MA Bosniak 1986 [33]; the 2005 update by GM Israel and MA Bosniak [34]; the version by SG Silverman et al., 2019, with consideration of MRI [30]; and the EFSUMB CEUS proposal by V Cantisani et al., 2021 [32].

## 2. Bosniak Classification

### 2.1. What Has Been Achieved by the CT-Based Bosniak Classification?

#### Are There Evidence-Based Imaging Criteria for a Risk Assessment of Renal Cystic Lesions?

In radiological imaging, irregular septa, thickness of more 2 mm, nodular changes, irregular wall thickening, and significant contrast enhancement indicate malignancy [35,36]. Calcification is a non-specific sign [37]. In contrast-enhanced CT, the presence of nodular or septal enhancement showed the highest sensitivity for predicting malignancy with moderate to good interobserver agreement. All cystic renal cell carcinomas had an enhanced septal or nodular component. The mean sensitivity and specificity in predicting malignancy for the presence of septal enhancement were 83% and 82%, respectively; for nodular enhancement, 67% and 96%; and for septal or nodular enhancement, 100% and 86%, respectively [38].

In a CEUS study [24] with histopathologically confirmed Bosniak III cysts, 62% had septa, of which 61% were malignant. Again, 75% of all focal thickened and contrast enhanced septa were malignant [24].

According to the 2019 Bosniak classification, cystic masses with many or thick enhancing septa are associated with a higher likelihood of malignancy than masses with few or thin septa [30]. The total volume of enhancing tissue in a cystic mass may be the most important indicator of whether that mass is benign or malignant [30].

In a pathologic study [20], mural nodules were associated with a higher probability that the cystic masses were malignant (*p* = 0.01). The presence of a mural nodule was a strong predictor of malignancy. There was also a trend toward a higher risk of malignancy in cysts with larger solid components (*p* = 0.07). However, there were no significant associations between other radiologic parameters and the likelihood of malignancy [20]. Cystic lesions without septations were more likely to represent papillary RCC compared with those with septations (*p* < 0.01) [20].

In a recent systematic review and meta-analysis of retrospective studies, the application of CECT, CEMRI, and CEUS in the evaluation of renal cystic lesions was analyzed [39]. A total of ten relevant articles were included in this meta-analysis. The results of this meta-analysis indicated that CEUS had high sensitivity and specificity in diagnosing renal cystic lesions, which was statistically significant.

### 2.2. If a Cystic Renal Lesion Is Detected with Primary Imaging (e.g., US): When Is Further Examination Necessary to Assess the Risk of Malignancy Indicated?

Bosniak cysts class I and II are classified as benign; Bosniak class III as indeterminate, with a malignant potential of about 50% [40]; and class IV as predominantly malignant. The category of IIF was introduced to differentiate and progress those inconclusive cysts of category II or III [35]. IIF indicates “follow-up”. Thus, less benign cysts should be subjected to unnecessary surgical resection with associated risks.

In a meta-analysis [41], before the introduction of the IIF category, the proportion of malignancy in the different categories was as follows: in Bosniak category I, 1.7%; in category II, 18.5%; in category III, 33.0%; and in category IV, 92.5%.

In a later meta-analysis [42], the percentage of malignant cysts in category I was 0%; in category II, 15.6%; in category IIF, 0%; in category III, 65.3%; and in category IV, 91.7%. 

Two of the studies involving malignant cysts in category II had no category IIF classified [36,43]. 

A third study on malignant cysts in category II classified the category of IIF, but there were no malignant cysts in this category. In class IIF, only three patients were included (2.9%) [44]. It must be considered that the update of the Bosniak classification 2005 [34] in the IIF class also included non-enhancing high-attenuation renal lesions > 30 mm. Regarding the size, there was no information on the sizes of these 2.9%.

Of further importance is the ISUP (International Society of Urological Pathology) grading system. This is used to predict the biological aggressiveness of the cancer. An ISUP grade of 1 or 2 is indicative of a relatively indolent cancer. Of all resected cysts (Bosniak ≥ IIF), 73.8% had a low ISUP stage [45].

The progression rates of Bosniak IIF cysts have been reported from radiological imaging and CEUS to be 4.6% [45,46]; 7.1% [47]; 10.9% [48]; 12% [40]; 13% [49]; and 14.8% [50]. The malignancy rates of Bosniak II F cysts that progressed and then underwent surgery were 60% [45]; 86% [40]; 87.5% [49]; and 100% [46].

Progression from Bosniak IIF to Bosniak III or IV usually occurs within the first two years of follow-up [49]. If a Bosniak IIF cystic mass shows progression, then the likelihood of malignancy is high. In contrast, the malignancy rate of IIF cysts without progression was 1% [40].

The malignancy potential of Bosniak III cysts was reported as 50% in the 2019 update of the Bosniak classification according to the meta-analysis of Schoots et al. [40]. This indicates that cystic masses are also benign, and in case of malignancy, the tumors are not aggressive [30]. The malignancy rate according to biopsy was 60.7% [51], and in surgically resected Bosniak III cysts, 50% [49], 60% [52], 66% [24], and 81.8% [50]. The approach to Bosniak III cystic lesions has shifted toward surveillance in recent years [30]. Metastatic disease at primary diagnosis is rare, and comprised only 2.2% of cases [20]. 

A survey of Canadian urologists evaluated their approaches to Bosniak III and IV cysts [53]. 

Of the urologists, 13.7% never or rarely (<5% of cases) offered active surveillance for Bosniak III, whereas 33.1% provided active surveillance in >50% of patients with Bosniak III cysts for whom surgical excision was an appropriate treatment option. Regarding Bosniak IV cysts, a majority of urologists (60.1%) offered no or rare (<5% of cases) active surveillance of Bosniak IV cysts, whereas only 10.1% offered active surveillance in >50% of cases [53]. Surveillance of Bosniak III cysts was considered by significantly more urologists from academic centers than from non-academic centers [53].

The American Urological Association guidelines recommended surgery for Bosniak III and Bosniak IV masses 2 cm or larger in patients whose life expectancy is not limited, despite interest in surveillance [54,55]. However, the 2021 update recommends active surveillance with potentially later intervention. This applies to patients with solid, enlarging renal masses <2 cm or Bosniak 3–4 lesions that are predominantly cystic. Shared decision making about active surveillance should weigh the risks of intervention or competing mortality against the potential oncologic benefits of intervention. Recommendations for biopsy of renal masses and considerations for regular clinical/imaging-based surveillance are discussed [56].

### 2.3. Description of the Original Bosniak Classification and of Different Versions; Consequences for Patient Management

The first classification of renal cysts by Bosniak into classes I/II/III/IV dates back to 1986 [33] [Table 1]. There have been several revisions. In 2005, the IIF class was introduced [34,41] [Table 2]. In 2019, an update was developed that officially included MRI [30] [Table 2]. For the CEUS, a proposal was developed in 2020 by the EFSUMB [32] [Table 3].

### 2.4. What Are the Detailed Issues Arising from the Bosniak Variation 2019 and the EFSUMB-CEUS Proposal?

When comparing the classifications according to Silverman [30] and the EFSUMB-CEUS proposal according to Cantisani [32], some questions arise. 

A type of Bosniak class II is the presence of up to three thin septa ≤ 2 mm. A possible enhancement is described for CT and MRI [30]. The CEUS proposal generally describes a nonenhancement of the septa or only individual microbubbles in tiny vessels [32]. However, enhancement should not exclude class II if the B-mode criteria of max. three septa with thicknesses of ≤2 mm is not exceeded. In the authors’ experience, enhancement may be visible in a septum with a maximum thickness of 2 mm. 

The same applies to the IIF class. While septal enhancement may be present in CT and MRI [30], nonenhancement or only individual microbubbles running within tiny vessels are described in the CEUS proposal [32]. Here, too, enhancement of the septa should be permitted for Class IIF if the other B-mode criteria correspond to Class IIF. As the CEUS proposal from 2020 only describes an enhancement of the septa in class III, this could lead to difficult interpretations. The line between individual microbubbles and enhancing septa can be very narrow as long as there are no more objective measurement parameters.

A question of classification arises for cysts with septa that are >3 mm and <4 mm thick. Are these assigned to class IIF or III? Both the Bosniak Classification 2019 [30] and the EFSUMB-CEUS proposal [32] describe a septum thickness of up to 3 mm for class IIF, but septa ≥ 4 mm for class III. 

### 2.5. Accuracy and Interobserver Agreement of CT in Comparison to MRI for Cystic Renal Lesions

#### 2.5.1. How Accurate Is the Risk Assessment of Renal Cystic Lesions Using the CT-Based Bosniak Classification?

It is important to distinguish the categories of IIF and III. However, a meta-analysis [40] according to the Bosniak classification criteria prior to the 2019 update to assess interreader agreement showed that the reported k values were largely due to agreement in Bosniak I and IV masses. The absolute discrepancy ranged from 6% to 75%, and was particularly pronounced for Bosniak II, IIF, and III masses [30,40].

In a CT study with two radiologists, there was good agreement for all renal cysts >/= class II. The linear weighted kappa value was 0.69 for all complex cysts. However, the greatest disagreement was for class II cysts at 47%, 20% for IIF, 28% for III, and 28% for IV. Disagreement was significantly higher for Bosniak II cysts than ≥IIF cysts (*p* = 0.01). Bosniak II and IIF cysts were initially overclassified by the initial reader in 46% and 10% of cases, respectively. Downgrading was performed by an uroradiologist [45].

A further meta-analysis was performed to evaluate the diagnostic performance of the 2019 Bosniak classification [57]. In the Bosniak Classification 2019, previously subjective descriptions were precisely defined [30]: “thin” is now defined as ≤2 mm, “minimally thick” as 3 mm, and “thick” as ≥4 mm; “few” is defined as 1–3 septa and “many” as ≥4 septa. Therefore, each node was a kind of convex protrusion arising from a wall, and septa were either nodules (any size if margins are acute with walls or septa), 4 mm if obtuse margins with wall or septa) or irregular thickening (3 mm if margins are obtuse with wall or septa) [30].

The sensitivity and specificity results of eight studies on CT and/or MRI [58,59,60,61,62,63,64,65] were 0.85 (95% CI 0.79–0.90) and 0.68 (95% CI 0.58–0.76), respectively [57]. However, the results were highly heterogeneous. For individual studies, the sensitivity ranged from 0.76 to 1.00, with a specificity of 0.40–0.84 [57]. The 2019 version demonstrated significantly higher specificity (0.62 vs. 0.41, *p* < 0.001); however, with a significant decrease in sensitivity (0.88 vs. 0.94, *p* = 0.001) [57].

Regarding the four studies [58,60,62,65] reporting the diagnostic accuracy of CT, the pooled sensitivity and specificity were 0.86 (95% CI 0.68–0.95) and 0.71 (95% CI 0.61–0.80), respectively [57].

#### 2.5.2. How Accurate Is the Risk Assessment of Renal Cystic Lesions Using the MR-Based Bosniak Classification?

More category IIF lesions are seen on MRI than on CT. Thus, more benign cysts may be placed in category III or even operated on. This exaggerates the thickness of the septa. The enhancement is emphasized. These cysts are, therefore, classified higher upon MRI than CT. Bosniak emphasizes this especially for lesions smaller than 2.5 cm, where the intraluminal ones on MR imaging may appear more prominent and thicker. These putatively thicker septa occupy a larger percentage of the cyst. This problem was less observed in larger cysts with more fluid. While CT cannot differentiate intraluminal structures in hemorrhagic cysts with high signal intensity, MR images, with image subtraction, can show the wall thickness and the enhancing tissue inside the lesion. Thus, some hemorrhagic cystic lesions can be downgraded from category III to IIF or upgraded to category IV [35]. However, Bosniak himself considered the quality of MRI images to be highly variable, which also makes the classification of renal cysts difficult [35].

According to the meta-analysis [57] that evaluated the diagnostic performance of the 2019 variation of the Bosniak classification, in five studies [59,60,62,63,65] reporting the diagnostic accuracy of MRI, the pooled sensitivity was comparable (0.87, 95% CI 0.78–0.93), but the specificity was lower (0.67, 95% CI 0.48–0.81) [57]. In three studies [58,61,64] reporting the diagnostic accuracy of CT in combination with MRI, the pooled sensitivity (0.81, 95% CI 0.70–0.91) was lower than using CT or MRI alone, while the specificity was comparable (0.67, 95% CI 0.61–0.82) [57]. There was no significant difference in accuracy between CT and MRI [57].

### 2.6. How Good Is the Interobserver Agreement in CT- and MRI-Based Classification of Renal Cystic Lesions? 

Interobserver agreement for septal and nodular enhancement on contrast-enhanced CT was good (kappa = 0.67) and moderate (kappa = 0.57), respectively [38].

In the meta-analysis [57] considering risk stratification of renal cysts on MRI according to the Bosniak 2019 classification, the majority of studies did not yield a significant improvement in inter-reader radiological agreement. The update of the Bosniak classification did not appear to have resolved the previously identified problems with variability between radiological readers. 

In addition, a study found that inter-reader agreement of the Bosniak Classification, Version 2019, was lower than that of the 2005 version for both CT and MRI [66]. The findings of a total of eighteen residents and fellows were compared. The best agreements were in categories I (κ = 0.49–0.69) and IV (κ = 0.45–0.51). The poorest agreements occurred in the important category, IIF (κ = 0.18). The Bosniak classification 2019, when compared to the Bosniak classification 2005, did not improve interobserver agreement nor diminish the proportion of masses categorized into lower Bosniak classes among non-subspecialized readers [66]. 

In an MRI study comparing the interobserver agreement for risk stratification of renal cysts, only moderate agreement was found among two subspecialized investigators—for the Bosniak classification 2005, kappa = 0.57, and for the 2019update—kappa = 0.55 [63]. According to these data, the main goal of improved risk stratification with the 2019 update was not achieved.

In another MRI study [59] interobserver agreement was higher with version 2019 than version 2005 (weighted k = 0.64 vs. 0.50, respectively). However, interobserver agreement between senior and junior radiologists did not differ between the 2019 version (weighted k = 0.65 vs. 0.64, respectively) and the 2005 version (weighted k = 0.54 vs. 0.46) [59]. But the diagnostic specificity for malignancy was higher with version 2019 than with version 2005 (83% vs. 68%), respectively, without any difference in sensitivity (89% vs. 84%) [59].

However, with specialization in urological radiology and many years of professional experience, a near-perfect agreement (0.85) can be obtained [67].

## 3. Accuracy and Interobserver Agreement of CEUS in Comparison to CECT for Cystic Renal Lesions

### 3.1. What Is the Accuracy of a CEUS-Based Bosniak Classification Compared to the CT-Based Bosniak Classification and/or Compared to Clinical Outcome Data? 

CEUS shows a higher overall accuracy compared to CT: a sensitivity of 100 vs. 73.7%, a specificity of 81.4 vs. 83.7%, a positive predictive value of 70.4 vs. 66.7%, and a negative predictive value of 100 vs. 87.8% [68]. 

CEUS detected more intracystic septa and a thicker wall, as well as more enhancement and solid nodules compared to CECT, which could not be delineated on CT [69]. More thin septa in renal cysts were diagnosed; therefore, more cysts were classified as IIF in CEUS than in CT [70]. The concordance between CEUS and CT in the assessment of vascularization was high (0.77, *p* < 0.001). However, CEUS had greater resolution with respect to the finest vessels in the septa. CEUS is more sensitive to contrast enhancement than CT [70].

There was complete agreement on CEUS and CECT in terms of differentiating between cysts that should and should not be operated on [70].

Among histologically confirmed malignant cysts, the diagnostic efficiency of CEUS was 90% versus 74% on CT [69,71]. 

CEUS has an adequate diagnostic performance in comparison to CT and MRI [47,68,70,72,73,74,75,76]. In a meta-analysis of 1142 renal cystic masses, the pooled sensitivity, specificity, positive likelihood ratio, and negative likelihood ratio for CEUS/MRI were 0.95/0.92, 0.84/0.91, 5.62/6.74, and 0.09/0.13, respectively [73].

The high resolution and detection of sensitive visualization of vascularization may result in higher CEUS Bosniak classes than CT/MRI Bosniak classes (40%) [52]. For this reason, the agreement may be lower. In one study, CEUS Bosniak classes matched CT/MRI Bosniak types in only 58% of cases (κ = 0.28). However, this is more a reflection of the superior resolution of CEUS in imaging vascularization [52].

### 3.2. Is Hyperattenuation Correlated with Any Ultrasound Criterion (Bosniak II)?

Hyperattenuation on CT occurs in the visualization of proteinaceous fluid or densely packed cells. These are usually benign hemorrhagic or proteinaceous cysts [77]. Silverman characterized benign cysts with hyperattenuation as homogeneous and 70 HU or greater on native CT [30]. On CE-CT, according to renal protocol, these are then non-contrast-enhancing, homogeneous, and greater than 20 HU [30]. Hyperattenuated cysts may have internal echoes on ultrasound. This can make it difficult to differentiate solid lesions on ultrasound [77]. Other lesions that impress as hyperattenuating lesions are hematomas, vascular malformations, aneurysms, and inflammatory processes [77]. Aneurysms and vascular malformations can be detected by CDI, although they are not always differentiated. Hematomas and focal inflammatory processes can be diagnosed by CEUS in conjunction with clinical appearance. If necessary, vessels and, thus, solid tissue can be detected with color Doppler imaging. For a reliable differentiation, a contrast-enhanced procedure, in this case CEUS, is required. Solid hyperattenuating renal masses can have both malignant and benign causes. Possible malignant causes are renal cell carcinoma and lymphoma; benign causes are angiomyolipoma with minimal fat [77]. 

### 3.3. How Good Is the Interobserver Agreement in CEUS-Based Classification of Renal Cystic Lesions? 

In the study by Ascenti et al., the interobserver agreement was high (0.86, *p* < 0.001) for the classification of renal cysts with CEUS [70]. More studies are needed in order to assess interobserver agreement in CEUS. In our own experience, the evaluation of intracystic septa requires precise characteristics and specifications regarding the number, as well as measurement of focal thickenings, in order to classify them. It is likely that a more precise specification of the size of contrast-enhancing nodules would influence the interobserver agreement. Also, in CEUS, the results are likely to depend on the quality of the investigator’s training and experience. 

## 4. How to Perform US and CEUS in Cystic Renal Lesions and Reporting of Results

### 4.1. Which Features of Renal Cystic Lesions Should Be Described in an Ultrasound Report in Order to Allow for Classification?

The location and size of the cyst and its structural characteristics are basic data of every ultrasound finding. Is it a cyst with typical cyst criteria (thin wall < 2 mm [32] without irregularities, entry echo, exit echo, anechoic content, dorsal acoustic enhancement, and cyst margin or tangential shadow). If an evaluation of the cyst wall finds the thickness to be <2 mm [32]; are there any irregularities?

Are there internal structures in the cyst, i.e., number and thickness of septa, solid mural nodules and their size, or debris? Are there calcifications? A structured reporting system can be particularly helpful for less experienced examiners [78].

### 4.2. Which Sonographic Criteria of Renal Cystic Lesions Indicate the Use of Contrast-Enhanced Imaging for Further Risk Assessment?

The irregularities and thicknesses (>2 mm [32]) of the cyst wall, internal structures (septa, mural nodules, lumen is not anechoic) should indicate contrast-enhanced imaging. The size of enhancing internal structures/nodules/potential tissues should be measured. 

Bosniak class I and II cysts do not require further contrast-enhanced investigation. Classes IIF, III, and IV need further contrast-enhanced investigation.

### 4.3. What Are the CEUS Criteria to Assign a Renal Cystic Lesion to the Bosniak Classes I–IV? (CEUS Adaptation(s) of Bosniak Classification)

The EFSUMB proposal 2020 of the CEUS adaptation of Bosniak classification 2020 is presented in Table 3.

### 4.4. If There Are Different CEUS-Adaptations of the Bosniak Classification: Which Should Be Recommended for Clinical Use?

The problem of overstaging in Bosniak classes IIF, III, and IV in CEUS requires verification. The inclusion of the contrast-receiving nodes according to size in classes IIF and III is an attempt at more accurate differentiation. Contrast-enhancing nodes are the one malignancy feature. In the CEUS-Bosniak classification (EFSUMB 2020), no contrast-accepting nodules are described in classes IIF and III. In the current proposal, class IIF now includes contrast-enhancing nodules ≤ 5 mm and class III > 5 mm ≤ 10 mm. Bosniak class IV no longer includes all contrast enhancing nodules, but only those >10 mm (<20 mm).

### 4.5. What Would Concretely Change by Considering the Node Size in the Classification of CEUS Classes?

Enhancing irregular (displaying </=3-mm obtusely margined convex protrusion(s)) walls or septa are described for class III [30]. According to Silverman, these are not explicitly defined as nodules, but as enhancing convex protrusions, which are irregular thickenings. But according to the current proposal, these protrusions could be moved to class IIF. Class IV has, so far, included enhancing convex protrusions/nodules with acute margins of any size [30]. According to the new CEUS proposal, the sizes of the nodules would differentiate the classes. Not every size would belong to class IV. Nodules ≤5 mm would correspond to class IIF, 5–≤10 mm to class III, and 10 mm–≤20 mm to class IV. Furthermore, Class IV enhancing convex protrusions/nodules with obtuse margins ≥4 mm would still belong [30]. These would also be assigned to the respective classes according to their sizes. Those ≤5 mm would still be defined as class IIF. The same applies to the EFSUMB-CEUS proposal regarding soft tissue protrusions/nodules with obtuse margins (≥4 mm) or with acute margins of any size [32].

### 4.6. Which Features of Renal Cystic Lesions Should Be Reported in a CEUS Report of Kidney Cyst Examination? 

The presence or absence of contrast enhancement of wall irregularities, septa, and mural nodules or tissues should be reported. The sizes of enhanced mural nodules and tissues should be measured. The size of the node is a major criterion for classification into classes IIF/III/IV [79].

### 4.7. Should a CEUS Report Include Information on Examination Conditions/Examination Quality/Limitations?

Certain examination conditions may limit CEUS similarly to US: poor visibility due to high body mass index, intestinal gas overlay, localization of a cyst away from the transducer, slice thickness artifacts, and acoustic cancellation due to calcifications in the cyst wall or lumen. 

These should be named, as incomplete visibility of the region of interest compromises accessibility and accuracy.

### 4.8. How Should We Perform CEUS to Classify Renal Cystic Lesions Practically? Which Contrast Media May/Should Be Used? Is TIC Analysis Useful for the Classification of Renal Cystic Lesions?

CEUS of cystic masses of the kidneys is performed according to the guidelines in [32,80,81,82]. CEUS in renal assessment is considered off-label use [81]. 

The EFSUMB Position Paper on a Bosniak-adapted renal cyst classification based on CEUS [32] recommends advanced ultrasound experience, according to EFSUMB competency level 2, as a prerequisite for performing CEUS examinations of the kidneys [32]. As shown for other applications, there is a clear association between the examiner’s experience and credibility of CEUS results [81,81,83,84]. A structured reporting program should be used.

A prerequisite for CEUS is a good setting of the corresponding region in the B-scan US. The ultrasound machine must be equipped with low-mechanical-index procedures (MI < 0.3). A US scanning machine must be optimized for CEUS acquisition and the post-processing of data. The amount of contrast agent used is based on the recommendations of the device manufacturer. Too small an amount of contrast agent may result in less sensitive visualization of vascularization. In contrast, too large a contrast agent dose can lead to excessive enhancement of structures. Usually, SonoVue^TM^ is used for examinations in most European countries. In other countries, the respective approved ultrasound contrast medium is used. SonoVue^TM^ has a very good safety profile [85,86]. There are not yet sufficient practice-relevant data available on the inclusion of dynamic contrast-enhanced ultrasound (DCE-US) [87] and TIC analysis for the diagnosis of malignant cysts. It would be interesting to derive curves in the areas of nodes.

### 4.9. Beyond Assessment of Malignancy Risk: Are There Further Indications of CEUS in the Evaluation of Cystic Renal Lesions (e.g., Suspicion of Bleeding, Suspicion of Infection, Guidance of Interventions)

CEUS can be used to visualize the finest vascularization. This also applies to the escape of microbubbles into the lumen of a cyst in the case of active bleeding. However, active recording of this event should be rare. In any case, active bleeding can be excluded with CEUS.

In the case of a cyst infection and corresponding clinical symptoms with fever and pain, renal cysts may show abscess-like changes on B-mode sonography and CEUS.

Percutaneous thermal ablation is a potential treatment option. Gas formation occurs during ablation, so CEUS should not be performed until 10–15 min after the intervention to estimate tumor necrosis. During follow-up, tumor recurrences can be detected by CEUS [81].

## 5. Comparative Issues

### 5.1. What Are the Advantages/Limitations/Pitfalls of CEUS-Based Bosniak Classification Compared to CT- and/or MR-Based Bosniak Classification? 

The advantages, limitations, and pitfalls of CEUS-based Bosniak classification compared to CT- and/or MR-based Bosniak classification are presented in Table 4.

### 5.2. Does CEUS-Based Classification Tend to Overrate or to Underrate the Risk of Malignancy of Renal Cystic Lesions?

In the study by Graumann et al. [90], CT was the gold standard compared to CEUS and MRI. 

There was 79% agreement between CT and CEUS and 78% agreement between CT and MRI. In this study, upgrades and downgrades by CEUS were present in approximately equal proportions. They concerned classes IIF–IV. Compared to MRI, CT predominantly downgraded the mismatched cases. Here, however, pathology was missing as the gold standard [90].

In a study with CEUS, CECT, and CEMRI by Lerchbaumer et al. [91], there was good agreement between CEUS and CECT, but differences between CEUS and MRI. There was an overall agreement of 73.6% between CEUS and CT. Most differences were found in the important classes IIF and III. A downgrade of 21.8% occurred in CT. The intraclass coefficient between CEUS and CT was 0.824 (95%-CI: 0.74–0.88; *p*-value < 0.001) [91]. There were greater differences between CEUS and MRI in all classes. A downgrade also occurred predominantly in MRI. The intraclass coefficient was correspondingly lower: 0.651 (95% CI: 0.51–0.76; *p* < 0.001) [91]. These results are somewhat contradictory to the description that MRI has a higher spatial resolution than CT [35]. A downgrade compared to CEUS would not be expected. However, one could conclude that additional details in the CEUS Bosniak classification would be helpful.

In the study by Herms et al. [52], CEUS/Bosniak classification matched with CT/MRI/Bosniak at 57.7%. Mismatches were noted in 42.3%. In 40.4%, CEUS-Bosniak resulted in higher stages than CT/MRI-Bosniak. Lower graduation was seen in only 1.9%. The higher resolution of CEUS can lead to overgrading in class III [52].

In a comparison of methods between CEUS, CECT, and CEMRI with the gold standard CT by Tshering Vogel et al. [92], CEUS and CT agreed by 51%. In 43%, the grading in CEUS was higher than in CT [92]. In the same study, MRI and CT showed a match of 80%. In 12%, the score was higher on MRI, and in 8%, lower than on CT [92]. These results confirm that renal cysts are graded higher than CT in case of discrepancy. Since there was no pathological backup, this remains only a comparison of methods.

### 5.3. What Are the (Comparative) Risks of Performing CEUS/CT/MRI in Renal Cystic Lesions?

Both MRI and CEUS have a higher detail resolution compared to CT. Thus, MRI and CEUS tend to overestimate Bosniak III cysts. There is a risk that potentially benign cysts may be either unnecessarily resected urologically or controlled by radiation-intensive CT. 

These documents might be worth citing when discussing the need to reduce radiation doses in medical imaging according to the FDA recommendations [93], American College Radiologists Image wisely CT [94], and the Royal College Radiologists UK. Cross-sectional imaging in cancer management is also used [95]. 

### 5.4. Should CEUS Be Considered an Equivalent Technique to Contrast-Enhanced CT in the Risk Assessment of Renal Cystic Lesions? (Or Should It Be Used Complimentarily to CT, or Alternatively, Only in Case of Contra-Indication or Non-Acceptance of Contrast-Enhanced CT?)

CEUS is an equivalent technique to contrast-enhanced CT in the risk assessment of renal cystic lesions. For complex cystic renal lesions on sonography, Herms et al. [52] suggest performing CEUS. In this study, the further procedure then depends on the CEUS–Bosniak stage. It is suggested that CEUS–Bosniak IIF cysts be sonographically controlled after 6 months. CEUS should be conducted if there is a relevant change in size or B-mode morphology. In the case of CEUS–Bosniak IIF cysts, this procedure eliminates the need for CT and MRI [52]. In case of primary CEUS–Bosniak III or IV cysts, the addition of CECT/MRI is recommended by Herms et al. [52]. CT and MRI are not utilized in stages III and IV. In the case of primary CEUS–Bosniak III or IV, the addition of CECT/MRI is recommended. In addition, discussion in an interdisciplinary clinical conference is given great importance. In the case of CEUS–Bosniak III, if CT/MRI shows only Bosniak IIF, sonographic control after 6 months is recommended, as for CEUS–Bosniak IIF. This minimizes overgrading by CEUS and reduces surgical overtherapy. In case of CEUS-Bosniak III/IV + CT/MRI </= III, the procedure is not clearly formulated. 

Here, additional equipment is required in order to select patients who actually have a Bosniak III stage cysts and, thus, an increased risk of carcinoma. The same applies to CEUS-Bosniak III/IV + CT/MRI Bosniak IV. However, in the latter case, malignancy is to be expected to a high degree, and surgery should be discussed in this situation [52]. 

In the Bosniak III category, according to all three imaging procedures, it would be desirable to develop additional criteria to specify the risk potential. The modified CEUS-Bosniak classification [79] presented here has the intention to contribute to more accurate scoring by including the sizes of contrast-enhancing nodes in classes IIF, III, and IV.

### 5.5. Is It Reasonable to Follow Up Renal Cystic Lesions Classified Primarily Using CT BOSNIAK Category I–IIF? 

Simple cysts that develop as a result of chronic kidney disease have a higher risk potential for malignancy [8]. For these, surveillance by B-mode sonography might be recommended. Surveillance is also conceivable in patients who have already been treated for renal cell carcinoma.

In some CT-based studies, there were malignant cysts in Bosniak classes I and II [41,42,44], although these classes are benign. This may have represented a downgrade by CT. It would be more effective to graduate Bosniak classes I and II in B-mode sonography and class II in CEUS than in CECT.

For Bosniak II F cysts, the follow-up applies. Bosniak [35] recommended control over 4 years; Lucocq et al. [45] every 6 months over 5 years. Herms et al. [52] recommended control by B-mode sonography after 6 months. Based on the knowledge that changes usually occur in the first 2 years [45], these controls should not be limited to 6 months. It would be conceivable to extend the controls to 5 years according to the recommendations of Lucocq et al. [45], but to perform B-mode sonography as in Herms et al. [52] and to perform CEUS in cases of changes in morphology or size. 

Rübenthaler et al. [47] observed progression in only 7.1% of IIF cysts on CEUS. This occurred after 12.9 months (SD ± 12.7 months), and 92.9% of the IIF cysts remain stable. In these cases, no CT or MRI would need to be performed [47].

### 5.6. In Which Cases Should CEUS-Based Classification Be Supplemented by CT- or MR-Based Classification?

The CEUS–Bosniak classification is based on the features of the Bosniak version with inclusion of CT and MRI. These are the same features that are assessed for scoring. The assessment of hyperattenuation in class II cysts is not necessary, as class II cysts can be reliably classified in CEUS.

### 5.7. Are There Criteria to Assign Patients with Cystic Renal Lesions to Either CEUS, Contrast-Enhanced CT, or MRI for Risk Assessment?

If the ultrasound conditions are good, Bosniak I-IIF cysts can be monitored by sonography; class II cysts with heterogeneous contents and IIF cysts can be evaluated with CEUS and monitored during the course. Type III lesions may be operated on in certain cases, e.g., young patients insisting on surgery. MRI might be considered. CECT is not indicated for renal indications, but perhaps for preoperative thoracic staging. Type IV lesions require excision or can be treated with ablation therapy. MRI is indicated. CECT of the abdomen and thorax should be performed preoperatively for staging purposes [79].

### 5.8. For Cystic Renal Lesions of CEUS–BOSNIAK Types I–IIF: Are There Special Follow-Up Recommendations Other than Those Given by Guidelines for CT-Bosniak Types I-IIF?

Morton A. Bosniak himself wrote that the length of follow-up of II F cysts is not entirely clear [35]. IIF cysts, again, are not a homogeneous group. There are those with minimal findings and those that are more complex. From his experience, he recommended a follow-up of 2 years for the less complex IIF cysts and 3–4 years for the more complex II F cysts. Progression usually occurs in the first few years [35]. In a CT study [45] in which 4.6% of II F cysts showed progression, this occurred within the first two years of observation, with median follow-up after a duration of 15.5 months (range, 5.6–35.1 months). Patients with Bosniak IIF cysts were followed with serial imaging at 6,12, 24, 26, and 48 months [45]. This represents a total of five CE-CTs with corresponding radiation exposure (!) of an abdominal CT, even in those patients in whom no progression or even regression were recorded. 

According to the proposal for a new CEUS-Bosniak classification [79] for Bosniak lesions of category IIF, CEUS follow-up is recommended yearly for 5 years. CECT and CEMRI are not necessary. Bosniak lesions in category II are benign cysts; CEUS is initially recommended, but does not require further follow-up contrast-enhanced imaging or intervention. CECT and CEMRI are not necessary. General follow-up is not recommended. Bosniak lesions in category I are benign cysts and do not require any further imaging, including CEUS, follow-up, or intervention. CECT and CEMRI are not necessary. Follow-up with US is also not necessary, with the exception of patients who are at an increased risk [79].

## 6. Conclusions

Compared to CT/MRI, CEUS is an equivalent procedure for the initial grading of renal cysts and the follow-up of CEUS Bosniak IIF and III cysts (Class IV should be treated). Due to its excellent resolution, the smallest vascularized structures can be made visible. To avoid overestimation and to resect potentially benign cysts, further refinement of stages IIF–IV is recommended. This could be carried out by differentiating the sizes of contrast-enhanced nodes in the cysts. Further studies are needed in order to evaluate this approach. In any case, CEUS is excellently suited for the follow-up of class IIF and III cysts. Why should predominantly benign cysts be repeatedly examined by CT over a period of years using a radiation-intensive procedure, when this is also possible using a less expensive procedure with fewer side effects? MRI can be used for class III cysts, and is recommended in class IV before surgery. CT is used more for staging before surgery.

To meet high diagnostic and reporting standards, the examiner must be appropriately qualified in CEUS. Structured report templates can also be helpful in ensuring quality.

## Figures and Tables

**Table 1 cancers-15-04709-t001:** Initial first version of the Bosniak Classification from 1986 [55].

Bosniak Class	Appearance
I	Uncomplicated, simple, and benign cysts
II	Minimally complicated cysts that are benign. These lesions include septated cysts, minimally calcified cysts, infected cysts, and high-density cysts.
III	More complicated cystic lesions. These lesions exhibit some findings seen in malignant lesions, and radiologically they cannot confidently be distinguished from malignant ones. All of these cases should be explored surgically unless clinically contraindicated.
IV	Clearly malignant lesions with large cystic components. These lesions show irregularity of margins and have solid vascular elements; while they are superficially cyst like, they are clearly malignant and should be treated by removal.

**Table 2 cancers-15-04709-t002:** Bosniak CT classification, 2005 [42], and proposed Bosniak CT/MRI, 2019 version [30].

Bosniak Class	Bosniak Classification 2005 [34]	Bosniak Classification Version 2019 [30]		Consequences [30]
	*CT Criteria*	*CT Criteria*	*MRI Criteria*	
I	Hairline-thin wall; water attenuation; no septa, calcifications, or solid components; non-enhancing	Well-defined, thin (≤2 mm), smooth wall; homogeneous simple fluid (29 to 20 HU); no septa or calcifications; the wall may enhance	Well-defined, thin (≤2 mm) smooth wall; homogeneous simple fluid (signal intensity similar to CSF); no septa or calcifications; the wall may enhance	No need further imaging and no follow-up
II	Few thin hairline septa in which “perceived” enhancement may be present. Fine calcification or a short segment of slightly thickened calcification in the wall or septa may be present. Uniformly high attenuation lesions < 30 mm (so-called high-density cysts) that are well marginated and do not enhance are included in this group.	***Six types***, all well-defined with thin (≤2 mm), smooth walls: Cystic masses with thin (≤2 mm) and few (1–3) septa; septa and wall may enhance; may have calcification of any type Homogeneous hyperattenuating (≥70 HU) masses at noncontrast CT Homogeneous non-enhancing masses. >20 HU on renal mass protocol CT, may have calcification of any type Homogeneous masses 9 to 20 HU on noncontrast CT Homogeneous masses 21 to 30 HU on portal venous phase CT Homogeneous low-attenuation masses that are too small to characterize	***Three types***, all well-defined with thin (≤2 mm) smooth walls: Cystic masses with thin (≤2 mm) and few (1–3) enhancing septa; any non-enhancing septa; may have calcification of any type Homogeneous masses markedly hyperintense at T2-weighted imaging (similar to CSF) on noncontrast MRI Homogeneous masses markedly hyperintense on T1-weighted imaging (approximately 2.5 times normal renal parenchymal signal intensity) on noncontrast MRI	“Benign Bosniak II renal cyst requiring no follow-up.”
II F	Multiple thin hairline septa or minimal smooth thickening of the wall or septa. Perceived enhancement of their septa or wall may be present. Calcification in the wall or septa that may be thick and nodular, but without measurable contrast enhancement. These lesions are generally well marginated. Intrarenal non-enhancing high-attenuation renal lesions > 30 mm are also included in IIF category.	Cystic masses with a smooth, minimally thickened (3 mm) enhancing wall; a smooth, minimal thickening (3 mm) of one or more enhancing septa; or many (≥4 mm) smooth and thin (≤2 mm) enhancing septa	***Two types***: Cystic masses with a smooth, minimally thickened (3 mm) enhancing wall; smooth, minimal thickening (3 mm) of one or more enhancing septa; or many (≥4 mm) smooth, thin (≤2 mm) enhancing septa Cystic masses that are heterogeneously hyperintense on unenhanced fat-saturated T1-weighted imaging	The large majority of Bosniak IIF masses are benign. When malignant, nearly all are indolent. Generally, Bosniak IIF masses are followed by imaging at 6 months and 12 months, then annually for a total of 5 years to assess them for morphologic change.
III	Thickened or irregular walls or septa with measurable enhancement	One or more enhancing thick (≥4 mm width) or enhancing irregular (displaying ≤3-mm obtusely margined convex protrusion(s)) walls or septa	One or more enhancing thick (≥4 mm width) or enhancing irregular (displaying ≤3 mm obtusely margined convex protrusion) walls or septa	Intermediate probability of being malignant, consider urology consultation.
IV	Soft-tissue components (i.e., nodule(s)) with measurable enhancement	One or more enhancing nodule(s) (≥4-mm convex protrusion with obtuse margins, or a convex protrusion of any size that has acute margins)	One or more enhancing nodule(s) (≥4-mm convex protrusion with obtuse margins, or a convex protrusion of any size that has acute margins)	The large majority are malignant, consider urology consultation.

**Table 3 cancers-15-04709-t003:** Multiparametric US and CEUS of renal cysts according to the Bosniak classification, EFSUMB 2020 Proposal for a Contrast-Enhanced Ultrasound-Adapted Bosniak Cyst Categorization [32].

Bosniak Class	B-Mode US	CEUS	Consequences
I	Simple cysts; thin wall (<2 mm); sharp margins without irregularities; no calcifications, anechoic content, or posterior acoustic enhancement	Thin wall without irregularities, no enhancement on CEUS or individual microbubbles running within tiny vessels in the wall	CEUS not necessary
II	Cysts that otherwise meet the criteria of simple cysts but are characterized by 1–3 thin septa (<2 mm) without irregularities. Calcifications of the wall and/or septa may be present which do not hamper evaluation of the cystic content Cysts with internal debris, echogenic content, or mixed appearance	Thin wall and septa without irregularities showing no enhancement, or individual microbubbles running within tiny vessels in the wall and septa Thin wall and septa without irregularities showing no enhancement, or individual microbubbles running within tiny vessels in the wall and septa	CEUS not necessary CEUS not necessary
IIF	Cysts with multiple septa, internal debris, echogenic content, or mixed appearance. Calcifications of the wall and/or septa may be present, slightly hampering the evaluation of the cyst wall, content, and septa Totally intrarenal cysts otherwise meeting the category II criteria	Multiple septa, thin or minimally thickened (2–3 mm). Smooth or minimally thickened wall Thin septa without irregularities may be present, showing no enhancement or individual microbubbles running within tiny vessels. Differentiation between non-enhancing and enhancing wall cannot be achieved	CEUS is recommended
III	Cysts with multiple septa, internal debris, echogenic content, or mixed appearance	Enhancing smooth, thick (≥4 mm) wall or septa, and/or enhancing irregular (>3 mm) walls and/or septa. No nodules are seen	CEUS is recommended
IV	Cysts with multiple septa, internal debris, echogenic content, or mixed appearance	Enhancing smooth, thick (≥4 mm) wall or septa, and/or enhancing irregular (>3 mm) walls and/or septa. Enhancing soft-tissue protrusions, either nodules with obtuse margins (≥4 mm) or with acute margins of any size	CEUS is recommended

**Table 4 cancers-15-04709-t004:** Advantages, limitations, and pitfalls of CEUS-based Bosniak classification compared to CT- and/or MR-based Bosniak classification.

	CEUS	CE-CT	CE-MRI
Limitations	Restricted sonographic examination conditions in obesity, intestinal gas overlay, cysts close to the transducer	Radiation exposure (especially during follow-up examinations); an abdominal CT has an exposure of 11 mSievert. High radiation exposure in repeated follow-up examinations of predominantly benign cysts	Restrictions on metallic medical devices and pacemakers
Contrast agent safety	High safety, independent of kidney function and thyroid function	Allergic reactions to iodinated contrast agents, dependent on kidney function, not in hyperthyroidism	Allergic reactions to gadolinium-based contrast agents: small risk of nephrogenic systemic fibrosis (NSF) in patients with advanced renal disease [88] Only sparse data regarding limited safety with respect to patients with acute kidney disease, risk factors for chronic kidney disease, and the risk of NSF among newer gadolinium-based contrast agents [89].
Weaknesses; strengths that can become weaknesses	Representation of the smallest vascularization structures; overestimation of the Bosniak classes **Therefore, the differentiation of the node size is newly added. Class III now includes nodes > 5 mm and ≤10 mm; and class IV includes nodes only >10 mm (<20 mm)**	High accuracy only with Bosniak I and IV Lower resolution compared to MRI and CEUS Poor differentiation of internal structures in Bosniak II cysts Underestimation of Bosniak classes	Higher resolution than CECT, overestimation of Bosniak classes

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
