# Peer review of "CEUS Bosniak Classification—Time for Differentiation and Change in Renal Cyst Surveillance"

_cancers, 2023, doi:10.3390/cancers15194709_

Round 1

Reviewer 1 Report

The authors need to revise the whole manuscript as reported. An organization for a narrative literature review should be as follows:

  • Introduction:
    • Introduce the topic of renal cysts and their clinical relevance.
    • State the review's objective, such as analyzing prevalence, risk factors, complications, and imaging options.
  • Methodology:
    • Describe the inclusion and exclusion criteria for considered studies.
    • Explain the sources used for literature search and the article selection process.
    • Discuss the approach to assessing the methodological quality of included studies.
  • Prevalence of Renal Cysts:
    • Examine prevalence data of renal cysts in detail, analyzing variations among different patient groups and imaging techniques.
    • Discuss possible reasons for prevalence variability.
  • Risk Factors:
    • Examine risk factors associated with the development of renal cystic lesions, analyzing evidence and clinical implications of each factor.
  • Risk of Malignancy:
    • Evaluate the risk of malignancy in cystic lesions, distinguishing between different lesion types and providing data on observed malignancy cases.
  • Types of Cystic Lesions:
    • Analyze different categories of renal cystic lesions, providing details about the characteristics of each category and discussing their clinical implications.
  • Symptoms, Complications, and Diagnosis:
    • Examine symptoms and complications associated with renal cystic lesions, as well as imaging techniques used for diagnosis.
  • Discussion:
    • Synthesize the main findings arising from the review.
    • Discuss the clinical implications of the results and future research directions.
  • Conclusions:
    • Summarize the main conclusions of the review and underscore practical implications for clinical management of renal cysts.
  • References:
    • Comprehensive lists of all sources cited in the review. Authors should read these novel papers on the topic to update their references and find a new point of discussion: PMID: 36363581; PMID: 37446024.

Organizing the review in this manner would enable a systematic exploration of various areas of interest, providing a more comprehensive and in-depth understanding of renal cysts and their clinical characteristics.

The authors need to revise the article and reorganize it in order to address the highlighted issues and enhance its overall structure and clarity.

Author Response

Introduction

Introduce the topic of renal cysts and their clinical relevance. State the review's objective, such as analyzing prevalence, risk factors, complications, and imaging options. Answer: We clarified the aim of the narrative review (page 3, line 100-105). “The aim of the analysis was to evaluate sensitivity, specificity and accuracy of CT, MRI and CEUS in the classification of renal cysts, the comparability of the imaging methods and the interobserver variability.” We also inserted in green color in the text: Renal cysts are common, especially in the elderly. Most of them are benign. It is not understandable why predominantly potentially benign lesions should be monitored by CT or MRI. In addition, CT is associated with radiation exposure. These methods should be reserved for truly complex findings. The focus of the present work is to review, based on objective criteria, when it is indicated to perform follow-up controls by CEUS.

We also included a new reference on a most recently published systematic review. “In a most recent systematic review and meta-analysis of retrospective studies the application of CECT, CEMRI, and CEUS in the evaluation of renal cystic lesions was analyzed (1). A total of ten relevant articles were included in this meta-analysis. The results of this meta-analysis indicated that CEUS had high sensitivity and specificity in diagnosing renal cystic lesions, which was statistically significant.” (page 3, line 133).

Methodology

Describe the inclusion and exclusion criteria for considered studies. Explain the sources used for literature search and the article selection process. Discuss the approach to assessing the methodological quality of included studies. Answer: We clarify that the submitted paper is not a systematic but a narrative review. Therefore, we did not use specific instruments like PRISMA 2020 or QUADS2 to systematically assess the methodological quality of included studies. We changed the text accordingly (page 3, line 100-105): “The aim of this analysis was to review and compare accuracy, interobserver variability, strengths and weaknesses of CT, MRI and CEUS in the classification of renal cysts. The analysis focuses on the classifications by MA Bosniak 1986 as original (2), the update 2005 by GM Israel and MA Bosniak (3), the version by SG Silverman et al 2019 with consideration of MRI (4) and the EFSUMB CEUS proposal by V Cantisani et al 2021 (5).

Other points

Prevalence of Renal Cysts: Examine prevalence data of renal cysts in detail, analyzing variations among different patient groups and imaging techniques. Discuss possible reasons for prevalence variability. Answer: We discussed as important factors different imaging techniques used and heterogenous study designs. The following points are marked in the manuscript by colors. Risk Factors: Examine risk factors associated with the development of renal cystic lesions, analyzing evidence and clinical implications of each factor. Risk factors in the text are marked with color. Risk of Malignancy (marked with colors): Evaluate the risk of malignancy in cystic lesions, distinguishing between different lesion types and providing data on observed malignancy cases. Types of Cystic Lesions (marked with colors): Analyze different categories of renal cystic lesions, providing details about the characteristics of each category and discussing their clinical implications. Symptoms, Complications, and Diagnosis: Examine symptoms and complications associated with renal cystic lesions, as well as imaging techniques used for diagnosis (marked with colors). Answer: The focus of the work is on the comparison of imaging techniques. The subject matter is covered in detail. However we inserted more informations underlying them in different colors corresponding to the single subparagraphs.

Discussion: Synthesize the main findings arising from the review. Discuss the clinical implications of the results and future research directions. We inserted in green-blue the main clinical implications and our proposal of clinical work-up changes. Conclusions: Summarize the main conclusions of the review and underscore practical implications for clinical management of renal cysts.

References

Comprehensive lists of all sources cited in the review. Authors should read these novel papers on the topic to update their references and find a new point of discussion: PMID: 36363581 (6); PMID: 37446024 (7). We thank  the reviewer for suggesting those papers that are very interesting but after carefully reading we found they deal with a different topic, “Micro-Ultrasound in the Diagnosis and Staging of Prostate and Bladder Cancer” (6) and “Urinary MicroRNAs as Biomarkers of Urological Cancers” (7) and therefore decided not to include them in our review. Instead and during revision we made a careful review of the literature. We inserted the new reference on the “Application of CECT, CEMRI, and contrast-enhanced ultrasonography in the evaluation of renal cystic lesions: a systematic review and meta-analysis of retrospective studies” (1).

Organizing the review in this manner would enable a systematic exploration of various areas of interest, providing a more comprehensive and in-depth understanding of renal cysts and their clinical characteristics. The authors need to revise the article and reorganize it in order to address the highlighted issues and enhance its overall structure and clarity. We showed the mentioned and structured content by colors.

Reviewer 2 Report

Major issue:

1. US assessment is considered higly subjective; I would expect that the Authors discuss the association between examiner's experience and credibility of CEUS results.

Minor issues:

2. In line 331: "Malignant possible malignant causes..." It is unclar.

3. In line 350 there should be comma instead of dot.

4. In line 351 there should be question marks instead of dots.

5. In line 374 ")" is lacking.

Author Response

Major issue:

US assessment is considered highly subjective; I would expect that the Authors discuss the association between examiner's experience and credibility of CEUS results. Response (we inserted- (page 11, line 422): The EFSUMB Position Paper on a Bosniak-adapted renal cyst classification based on CEUS (8) recommends advanced ultrasound experience according to EFSUMB competency level 2 as a prerequisite for performing CEUS examinations of the kidneys (5). As shown for other applications, there is a clear association between the examiner's experience and credibility of CEUS results (9-12). In addition, the text discusses that structured reporting programs should be used.

Minor issues:

  1. In line 331: "Malignant possible malignant causes..." It is unclar. “Malignant” was deleted. à Possible malignant causes are renal cell carcinoma and lymphoma; benign causes are angiomyolipoma with minimal fat.
  2. In line 350 there should be comma instead of dot. Done in the revised version.
  3. In line 351 there should be question marks instead of dots. Done in the revised version.
  4. In line 374 ")" is lacking. ")" inserted into the revised version.

Reviewer 3 Report

The primary objective of this review paper is to provide a comprehensive overview of recent data regarding the CEUS method, with the goal of promoting its more proactive integration into renal cyst monitoring strategies. Given the widespread adoption of the Bosniak classification by radiologists and urologists for evaluating renal cysts, the inclusion of the CEUS method within this classification is expected to yield improved outcomes.

The review is clear and comprehensive with relevance in the field. The gap in the knowledge was identified. There is no similar review published recently, so the present one is relevant and of interest for the scientific community.

The review comprises three well-organized tables that effectively illustrate the current state of the literature concerning the Bosniak classification and its 2019 revisions. It also provides a clear comparison between the CEUS-Bosniak classification and the CT/MRI-Bosniak classification.

The statements and conclusions are coherent and supported by the listed citations. There are no self-citations listed in the references.

This is an important review study which emphasizes that CEUS, an established technique, should be more prominently incorporated into renal cyst monitoring protocols. This review assesses the precision, advantages, and drawbacks of CEUS, CECT, and MRI for grading kidney cysts. To prevent potential overestimation with CEUS, it is necessary to introduce additional subcategories (IIF, III, and IV) and incorporate various size criteria for contrast-enhancing nodules. Enhancing the CEUS Bosniak classification through further refinement aims to better integrate CEUS into renal cyst monitoring and facilitate the development of sophisticated monitoring algorithms.

Lines 473-480: I recommend including the links in the references section.

Author Response

The primary objective of this review paper is to provide a comprehensive overview of recent data regarding the CEUS method, with the goal of promoting its more proactive integration into renal cyst monitoring strategies. Given the widespread adoption of the Bosniak classification by radiologists and urologists for evaluating renal cysts, the inclusion of the CEUS method within this classification is expected to yield improved outcomes. The review is clear and comprehensive with relevance in the field. The gap in the knowledge was identified. There is no similar review published recently, so the present one is relevant and of interest for the scientific community. The review comprises three well-organized tables that effectively illustrate the current state of the literature concerning the Bosniak classification and its 2019 revisions. It also provides a clear comparison between the CEUS-Bosniak classification and the CT/MRI-Bosniak classification. The statements and conclusions are coherent and supported by the listed citations. There are no self-citations listed in the references. This is an important review study which emphasizes that CEUS, an established technique, should be more prominently incorporated into renal cyst monitoring protocols. This review assesses the precision, advantages, and drawbacks of CEUS, CECT, and MRI for grading kidney cysts. To prevent potential overestimation with CEUS, it is necessary to introduce additional subcategories (IIF, III, and IV) and incorporate various size criteria for contrast-enhancing nodules. Enhancing the CEUS Bosniak classification through further refinement aims to better integrate CEUS into renal cyst monitoring and facilitate the development of sophisticated monitoring algorithms. We thank  the reviewer for his/her thorough and kind wording and review.

Lines 473-480: I recommend including the links in the references section. Done in the revised version. FDA recommendations (13), American College Radiologists Image wisely CT (14) and the Royal College Radiologists UK. Cross sectional imaging in cancer management (15):

References (cited in the response letter)

  1. Feng P, Yu L, Liang P. Application of CECT, CEMRI, and contrast-enhanced ultrasonography in the evaluation of renal cystic lesions: a systematic review and meta-analysis of retrospective studies. Biotechnol Genet Eng Rev 2023:1-14.
  2. Bosniak MA. The current radiological approach to renal cysts. Radiology 1986;158:1-10.
  3. Israel GM, Bosniak MA. An update of the Bosniak renal cyst classification system. Urology 2005;66:484-488.
  4. Silverman SG, Pedrosa I, Ellis JH, Hindman NM, Schieda N, Smith AD, Remer EM, et al. Bosniak Classification of Cystic Renal Masses, Version 2019: An Update Proposal and Needs Assessment. Radiology 2019;292:475-488.
  5. Cantisani V, Bertolotto M, Clevert DA, Correas JM, Drudi FM, Fischer T, Gilja OH, et al. EFSUMB 2020 Proposal for a Contrast-Enhanced Ultrasound-Adapted Bosniak Cyst Categorization - Position Statement. Ultraschall Med 2021;42:154-166.
  6. Calace FP, Napolitano L, Arcaniolo D, Stizzo M, Barone B, Crocetto F, Olivetta M, et al. Micro-Ultrasound in the Diagnosis and Staging of Prostate and Bladder Cancer: A Comprehensive Review. Medicina (Kaunas) 2022;58.
  7. Aveta A, Cilio S, Contieri R, Spena G, Napolitano L, Manfredi C, Franco A, et al. Urinary MicroRNAs as Biomarkers of Urological Cancers: A Systematic Review. Int J Mol Sci 2023;24.
  8. Cantisani V, Bertolotto M, Clevert DA, Correas JM, Drudi FM, Fischer T, Gilja OH, et al. EFSUMB 2020 Proposal for a Contrast-Enhanced Ultrasound-Adapted Bosniak Cyst Categorization - Position Statement. Ultraschall Med 2020.
  9. Sidhu PS, Cantisani V, Dietrich CF, Gilja OH, Saftoiu A, Bartels E, Bertolotto M, et al. The EFSUMB Guidelines and Recommendations for the Clinical Practice of Contrast-Enhanced Ultrasound (CEUS) in Non-Hepatic Applications: Update 2017 (Short Version). Ultraschall Med 2018;39:154-180.
  10. Sidhu PS, Cantisani V, Dietrich CF, Gilja OH, Saftoiu A, Bartels E, Bertolotto M, et al. The EFSUMB Guidelines and Recommendations for the Clinical Practice of Contrast-Enhanced Ultrasound (CEUS) in Non-Hepatic Applications: Update 2017 (Long Version). Ultraschall Med 2018;39:e2-e44.
  11. Rafailidis V, Huang DY, Yusuf GT, Sidhu PS. General principles and overview of vascular contrast-enhanced ultrasonography. Ultrasonography 2020;39:22-42.
  12. Minimum training requirements for the practice of Medical Ultrasound in Europe. Ultraschall Med 2010;31:426-427.
  13. White Paper: Initiative to Reduce Unnecessary Radiation Exposure from Medical Imaging. In. https://www.fda.gov/radiation-emitting-products/initiative-reduce-unnecessary-radiation-exposure-medical-imaging/white-paper-initiative-reduce-unnecessary-radiation-exposure-medical-imaging: Center for Devices and Radiological Health. US Food and Drug Administration; 2010.
  14. Image Wisely. A joint initiative of ACR, RSNA, ASRT and AAPM. In. https://www.imagewisely.org/Imaging-Modalities/Computed-Tomography: ACR, RSNA, ASRT and AAPM; 2009.
  15. Royal College Radiologists UK. Recommendations for cross sectional imaging in cancer management. Risks for radiation exposure. In. https://www.rcr.ac.uk/system/files/publication/field_publication_files/BFCR%2814%292_26_Risks.pdf: Royal College Radiologists UK; 2013.

Round 2

Reviewer 1 Report

the paper it is worthy of publication